# Revision Arthroscopic Bankart Repair: A Systematic Review of Clinical Outcomes

**DOI:** 10.3390/jcm9113418

**Published:** 2020-10-25

**Authors:** Chang-Jin Yon, Chul-Hyun Cho, Du-Han Kim

**Affiliations:** Department of Orthopedic Surgery, Keimyung University Dongsan Hospital, Keimyung University School of Medicine, Daegu 42601, Korea; poweryon88@gmail.com (C.-J.Y.); oscho5362@dsmc.or.kr (C.-H.C.)

**Keywords:** shoulder, Bankart, dislocation, instability, revision, arthroscopy, systematic review

## Abstract

Although the frequency of arthroscopic revision surgery is increasing in patients with recurrent dislocation after a primary shoulder stabilization, the literature describing arthroscopic revision Bankart repair has been limited. Preferred reporting items for systematic meta-analyses guidelines were followed by utilizing PubMed, EMBASE, Scopus, and Cochrane Library databases. Keywords included shoulder dislocation, anterior shoulder instability, revision surgery, stabilization, and arthroscopic Bankart repair. Quality assessments were performed with criteria from the methodological index for nonrandomized studies (MINORS). A total of 14 articles were included in this analysis. The mean MINORS score was 12.43. A total of 339 shoulders (337 patients) were included (281 males and 56 females). The mean follow-up period was 36.7 months. Primary surgeries were as follows: arthroscopic procedures (*n* = 172, 50.7%), open procedure (*n* = 87, 25.7%), and unknown (*n* = 80, 23.6%). The mean rate of recurrent instability after revision arthroscopic Bankart repair was 15.3% (*n* = 52), and an additional re-revision procedure was needed in 6.5% of cases (*n* = 22). Overall, there were 18.0% (*n* = 61) of complications reported. This systematic review suggests that arthroscopic revision Bankart repair can lead to an improvement in functional outcomes and reasonable patient satisfaction with proper patient selection.

## 1. Introduction

The arthroscopic treatment of anterior shoulder instability has become the preferred method for primary Bankart repair with reliable outcomes [1,2,3,4,5,6]. The advantages of arthroscopic repair for instability are the ability to accurately identify and treat the specific pathoanatomy, less iatrogenic damage to normal tissues, lower postoperative pain, and improved cosmesis. Some authors also report better functional recovery and improved motion compared with an open repair method [7]. Recently published studies show that arthroscopic procedures have similar outcomes and recurrence rates as open procedures [8,9,10]. Failure rates of initial arthroscopic stabilization procedures were reported to range from 5 to 15%, often needing additional revision surgery [11,12,13,14].

The treatment of recurrent instability after initial stabilization remains controversial. Traditionally, failures after initial treatment have been treated by open revision Bankart repair. However, surgeons have become more familiar with shoulder arthroscopy due to the evolution of surgical devices and increased educational opportunities, the use of arthroscopic revision surgery will also increase in the future.

To our knowledge, the current literature lacks a comprehensive review of revision arthroscopic Bankart repair for patients with recurrent instability after initial shoulder stabilization. Therefore, the purpose of this systematic review was to characterize outcomes following revision arthroscopic Bankart repair after failed initial anterior shoulder stabilization.

## 2. Materials and Methods

### 2.1. Search Strategy

PRISMA (preferred reporting items for systematic reviews and meta-analyses) guidelines were followed for the database search [15]. An extensive literature search was conducted on 2 December 2019, in PubMed, EMBASE, Scopus, and Cochrane Library databases. Using a Boolean strategy, the following field search terms were used: Search (shoulder) AND (Bankart OR dislocation OR instability OR stabilization) AND (revision OR reoperation) AND (arthroscopy OR “arthroscopic surgery”). The citations in the included studies were screened, and we also reviewed unpublished articles by conducting physical searches. The bibliographies of the relevant articles were subsequently cross-checked for articles that were not identified in the search.

### 2.2. Eligibility Criteria

Studies that met the following criteria were included: (1) English article, (2) full-text available, and (3) study on revision arthroscopic Bankart repair. Exclusion criteria were as follows: (1) non-English article, (2) full text not available, (3) articles on open revision surgery, (4) bony procedure in the initial surgery, and (5) no information on the pre- or postoperative clinical data. Articles that reported on clinical data for revision arthroscopic Bankart repair surgery after failed open shoulder instability surgery were also included.

### 2.3. Study Selection

Two reviewers independently reviewed studies returned from the initial database search. When a decision could not be reached for any particular article, that article was submitted to a third author for review and final decision. Throughout the duration of the search, the content of each article and its reference list were screened for overlap of patients from other studies.

### 2.4. Quality Assessment and Data Extraction

The level of evidence of the articles was collected. The methodological quality of the articles included in this meta-analysis was assessed using criteria from the methodological index for nonrandomized studies (MINORS), a validated tool to discern the methodological quality of nonrandomized studies. The highest possible score is 16 for noncomparative studies and 24 for comparative studies. Two blinded authors independently applied the MINORS and a final score was reached by consensus.

All study data were extracted with a standardized predetermined criterion form. The first author, published year, the number of groups in the study, type and design of the study, and level of evidence were extracted for the study characteristics. Mean age, gender ratio, dominant limb ratio, and mean duration of follow-up were extracted for the demographic data. Type of primary surgery, the interval from primary surgery until recurrence or revision surgery, preoperative factors of revision surgery, the average anchor used in revision surgery, concomitant procedures, and complication were extracted for the characteristics of surgery. Clinical outcomes were extracted with pre- and postoperative range of motion (ROM), and the measured clinical outcomes. A number of patients with shoulder re-instability after revision surgery, recurrence rate, the interval from revision surgery until re-recurrence, mechanism of failure, re-revision rate, type of re-revision surgery, and pathologic finding were extracted for the post-revision instability in the failure group.

## 3. Results

Inclusion and exclusion criteria were met by 14 articles [16,17,18,19,20,21,22,23,24,25,26,27,28,29]. The articles included in this study were published between 2002 and 2018. The flowchart of search, exclusion, and inclusion is included in Figure 1.

Included studies scored a mean 12.4 ± 3.2 (range 10–21) using the MINORS criteria. The detailed characteristics of the included studies are summarized in Table 1.

Demographic data were gathered and averaged when available. A total of 339 shoulders (337 patients) were included in the 14 studies (281 males and 56 females). The entire group had a mean age of 28.4 years (range 15–56 years). The mean follow-up was 36.7 months. All studies reported on arm dominance and 59.3% of the injured shoulders involved the dominant arm. The detailed demographic data of each included study are summarized in Table 2 and Appendix A.

Primary surgeries were as follows: arthroscopic procedures (*n* = 172, 50.7%), open procedures (*n* = 87, 25.7%), and unknown (*n* = 80, 23.6%). The exclusion criteria in the included articles were variable. Ten articles provided clear exclusion criteria for glenoid bone defects [16,19,20,21,22,23,24,26,27,29], and seven had exclusion criteria for humeral side bone defect [16,19,20,21,22,26,27]. Eight articles excluded patients with multidirectional instability for hyperlaxity [16,19,20,21,22,23,25,26], and the detailed data are summarized in Table 3.

Several pre-operative factors of revision arthroscopic Bankart repair were documented and included failure because of trauma (*n* = 110, 57.9%) and poor technique (*n* = 23, 27.4%). Glenoid bone loss was reported in 51 out of 144 shoulders (35.4%) in six studies [17,19,23,25,27,28]. Humeral bone loss was reported in 94 out of 146 shoulders (64.4%) in eight studies [17,22,23,24,25,26,27,28]. Pre-operative hyperlaxity was reported in 10 studies [16,17,18,19,22,23,26,27,28,29] (Appendix A).

Clinical range of motion of shoulder joint was reported in eight studies [16,20,21,22,23,24,25,26]; three of these reported only deficits compared with the contralateral side [20,21,23]. Although the reported range of motion in the included articles was variable, external rotation was the motion demonstrated to be statistically most affected by revision surgery. Pre- and postoperative patient-reported outcome scores for surgery were assessed in all included studies. Eleven outcome measures were reported in these studies, of which the most commonly used were the Rowe score (8 studies) [16,18,19,20,23,24,25,26], and Simple Shoulder Test score (6 studies) [17,18,19,21,23,26,29]. The aggregate mean preoperative Rowe score was 69.9 and postoperative was 85.3. Studies with Simple Shoulder Test scores reported postoperative improvement, with a mean preoperative of 7.1 and mean postoperative of 10.3. All other patients reported outcomes as presented in Appendix A.

Nine studies had a total of 102 shoulders (30.1%) receiving rotator interval closure [16,18,19,21,22,23,26,28,29], and three studies had a total of 28 shoulders (12.2%) receiving capsular plication as an adjunctive procedure [16,18,24]. The other concomitant procedures are described in Table 4. Overall, there were 61 (18.0%) complications reported. Instability recurrence, including positive apprehension sign and dislocation, was the largest portion (*n* = 52, 15.3%) of complications. The most common mechanism of failure after revision surgery was traumatic dislocation (27/52, 51.9%). In 22 shoulders, an additional revision procedure was required after the revision arthroscopic Bankart repair (Table 4).

## 4. Discussion

The important findings of this study are that although the rate of postoperative recurrence instability was 15.3% and the rate of total complications was 18.0%, revision arthroscopic Bankart repair yielded improved postoperative outcomes with high patient satisfaction at mid-term follow-up. In 2013, Abouali et al. reviewed 16 articles and 349 patients who underwent revision arthroscopic Bankart repair and reported a 12.7% rate of recurrent instability [30]. Comparatively, recurrent instability rates after arthroscopic primary Bankart repair for acute shoulder dislocation range from 7.7 to 19.6%; the revision rate was 7.1% [31]. Based on studies observed in more than ten years of follow-up after arthroscopic Bankart repair, the overall rate of recurrent instability was 31.2%, with 16.0% of patients having recurrent dislocations, and the overall revision rate was 17.0% [32].

The reason for the similar results to the primary arthroscopic Bankart repair is considered to be that most of the studies included in this systematic review had strict inclusion and exclusion criteria about bone loss. Bone defects caused by shoulder dislocation are reported to one of the most important contributing factors of failure after primary shoulder stabilization [12,33,34]. Among them, glenoid bone loss is reported to be an important risk factor for Bankart repair failure [33,34]. In a cadaveric study, an osseous defect at 3 o’clock with a width that was equal to or greater than 20% of the glenoid length significantly decreased anterior stability [35]. Shin et al. demonstrated a significant increase in instability of the shoulder, at more than 17.3%, in cases of the glenoid defect [36]. The engaging Hill-Sachs lesion has also been recognized as a risk factor for recurrent anterior shoulder instability [37,38]. In this review, 10 out of 14 studies presented exclusion criteria for glenoid bone loss ranging from 20 to 30%. Seven studies presented clear criteria for the Hill-Sach lesion. Four studies had no exclusion criteria relating to bone defects. However, a similar recurrence rate (17.5%) was observed in the group with no restriction on the bony lesion.

Various procedures emphasizing glenoid reconstitution have become the gold standard in patients with recurrent anterior shoulder instability associated with the bony glenoid defect. Among several surgical methods, the Latarjet procedure has been relatively successful for preventing recurrent dislocation and subluxation, particularly in patients with large glenoid bone loss or failed previous stabilization surgery [39,40]. In the long-term result, Hurley et al. reported the Latarjet procedure at a minimum of 10 years of follow-up resulted in excellent functional outcomes and a high rate of return to sport among athletes [41]. However, the Latarjet procedure is thought to have more complications, including infection, graft non-union, graft lysis, graft fracture, or neurovascular injury with arthroscopic shoulder stabilization procedures [42,43,44,45].

Residual capsular redundancy and plastic deformation may appear in patients with frequent shoulder dislocations or patients with general laxity. These pathologic conditions coexisting with Bankart lesions have been suggested as possible reasons for such recurrence, which indicates that Bankart repair alone might not be sufficient to restore shoulder stability [46]. In the systematic review by Randelli et al. [47], young age, male sex, and a competitive level of sports are patient-related factors associated with the recurrence of instability after Bankart repair. As arthroscopic techniques and instruments are developing, several technical factors have been proposed to increase the success rate of arthroscopic surgery in these patients.

Capsular shift or capsular plication can reduce the capsular volume and reinforce the redundant capsule, which may lead to improved glenohumeral stability and function [48]. Therefore, these techniques were recommended in high-risk patients with young or active collision sports [4,49]. Chiang et al. reported that patients treated with Bankart repair with posteroinferior capsular plication experienced a reasonable success rate (93.3%), and all patients without recurrence returned to their pre-injury levels of athletic activity at a minimum follow-up of 5 years [50]. Park et al., analyzed changes in the capsular volume of the glenohumeral joint with postoperative time after arthroscopic Bankart repair and capsular shift using a 3-dimensional image and reported that the total and anterior capsular volumes significantly decreased at 3-months postoperatively [51]. However, this reduction in capsular volume was not maintained at 1 year postoperatively. This group also found that female sex, being an elite athlete, and more dislocations before surgery were risk factors for increases in anterior capsular volume at 1 year.

Numerous clinical studies with rotator interval closure as adjunctive procedures have reported successful outcomes and suggested that indications for rotator interval closure be considered when the following conditions arise clinically: (1) anterior instability with a positive sulcus finding that persists in external rotation; (2) symptomatic instability and laxity in the inferior direction that does not disappear in external rotation with the arm at the side; (3) significant laxity and a large sulcus in the setting of multidirectional instability; and (4) patients with posterior instability that have an incompetent rotator interval [52,53]. In our study, capsular plication or rotator interval closure was performed as an adjunctive stability procedure in 130 (38.3%) shoulders in 10 studies. However, indications for adjunctive procedures were poorly reported, and the surgical techniques were inconsistent. Therefore, the establishment of surgical techniques and well-designed clinical studies are required with the usage of the capsular plication or rotator interval closure when combined with other procedures in the management of recurrent instability after primary surgery.

This systematic review has several limitations. First, most studies from which the data were extracted were low methodological quality, and the period of follow-up was relatively short (36.7 months). Second, there is heterogeneity among included patients in each study included in this review. For example, the inclusion and exclusion criteria, operative techniques, range of bone loss, and type of primary stabilization surgery are varied. Third, the initial stabilization surgeries varied among patients and studies.

## 5. Conclusions

This systematic review suggests that arthroscopic revision Bankart repair can frequently lead to an improvement in functional outcomes, low rate of reoperation, and high patient satisfaction with proper patient selection and surgical techniques. Above all, it is considered that the selection of the appropriate patient group is the most important. Therefore, well-designed clinical trials are warranted to determine the results of arthroscopic revision Bankart repair.

## Figures and Tables

**Figure 1 jcm-09-03418-f001:**
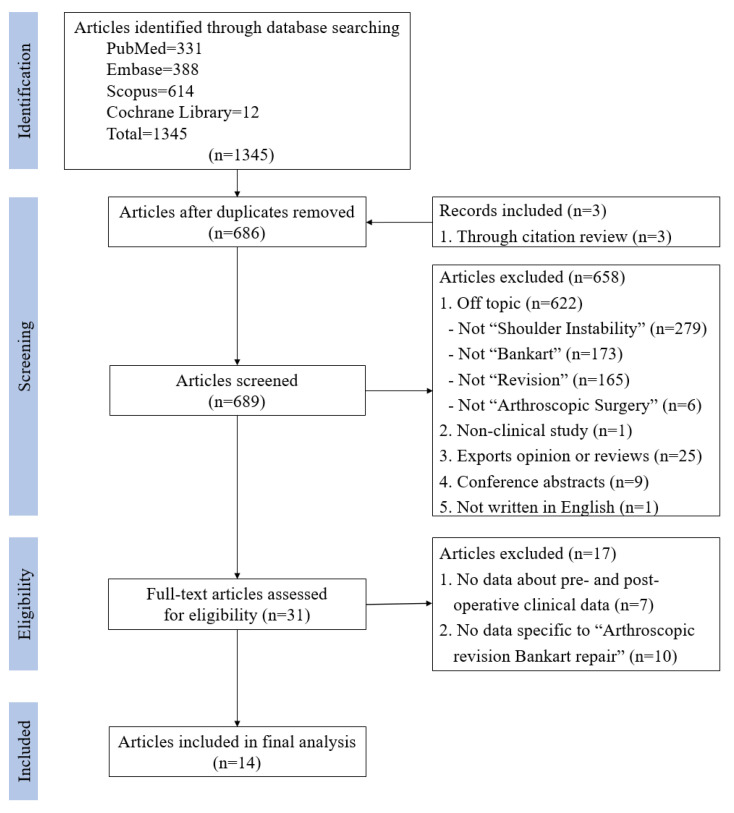
PRISMA (preferred reporting items for systematic meta-analyses) flow diagram.

**Table 1 jcm-09-03418-t001:** Study characteristics of included studies.

Authors	Year	Number of Shoulders	Type of Study	Design	Level of Evidence	MINORS Score
Arce et al. [16]	2012	16	Case series	Retrospective	IV	11/16
Balazs et al. [17]	2019	16	Cohort study	Prospective	II	14/16
Barnes et al. [18]	2009	17	Case series	Retrospective	IV	10/16
Bartl et al. [19]	2011	56	Case series	Retrospective	IV	11/16
Buckup et al. [20]	2018	23	Case series	Retrospective	IV	11/16
Creighton et al. [21]	2007	18	Case series	Retrospective	IV	11/16
Franceschi et al. [22]	2008	10	Case series	Retrospective	IV	12/16
Kim et al. [23]	2002	23	Prospective nonrandomized outcome study	Prospective	IV	12/16
Krueger et al. [24]	2011	20	Cohort study	Retrospective Comparative	III	21/24
Millar et al. [25]	2008	10	Case series	Retrospective Comparative	III	18/24
Neri et al. [26]	2007	12	Case series	Retrospective	IV	10/16
Patel et al. [27]	2008	40	Case series	Retrospective	IV	12/16
Ryu et al. [28]	2011	15	Case series	Retrospective	IV	11/16
Shin et al. [29]	2015	63	Case series	Retrospective	IV	10/16

MINOR, Methodological Index for Nonrandomized Studies.

**Table 2 jcm-09-03418-t002:** Patient demographics.

	Data
No. of patients in the study	337
No. of shoulders in the study	339
Mean age (years)	27.41
Gender	
Male	281 (83.4%)
Female	56 (16.6%)
Mean duration of follow-up (months)	36.7
Dominant extremity	201 (59.3%)

**Table 3 jcm-09-03418-t003:** Type of primary surgery and inclusion and exclusion criteria of revision surgery.

Authors	Primary Surgery	Interval from Initial Surgery to Recurrence or Revision Surgery (Months)		Exclusion Criteria of Revision Surgery		
		Initial Surgery to Recurrence	Between Procedure	Glenoid Loss	Humeral Loss	Hyperlaxity
Arce et al. [16]	Arthroscopic procedure: 10 Transglenoid: 4 Suture anchors: 6 Open procedure: 6 Staples + Open: 2 Transglenoid + Open: 2 Open: 2	NA	NA	> 25% excluded	> 33% excluded	Multidirectional instability excluded
Balazs et al. [17]	NA	NA	26.5 (10–62)	NA	NA	NA
Barnes et al. [18]	Arthroscopic procedure: 9 Arthroscopic thermal: 2 Arthroscopic suture plication: 1 Arthroscopic suture anchors: 5 2x Arthroscopic (unknown): 1 Open procedure: 8 Open Bankart: 5 Open Magnuson-Stack: 2 Open Magnuson-Stack & Bristow: 1	21.5 (4–83)	NA	No exclusion	No exclusion	NA
Bartl et al. [19]	Arthroscopic procedure: 34 Arthroscopic Bankart repair (anchors): 20 Arthroscopic Bankart repair (transglenoidal): 4 Arthroscopic Bankart repair (tacks): 8 Capsular plication/Capsular shrinkage: 2 Open procedure: 22 Open Bankart repair (anchors): 13 Open Bankart repair (transosseous): 6 Open capsular shift: 3	NA	43 (5–110)	> 20% excluded	Engaging Hill-Sachs excluded	Multidirectional instability excluded
Buckup et al. [20]	Arthroscopic Bankart repair: 23	28.7 ± 8.45	39.55 ± 31.62	> 20% excluded	> Calandra grade II excluded	More than 1° sulcus sign excluded
Creighton et al. [21]	Arthroscopic Bankart repair: 18 with thermal shrinkage: 9 without thermal shrirnage: 9	10 (4-20)	NA	> 25% excluded	Engaging Hill-Sachs excluded	Multidirectional or posterior instability Excluded
Franceschi et al. [22]	Arthroscopic Bankart repair: 10 Suture anchor: 9 Transglenoid: 1	18 (5-73)	25 (12-49)	> 30% excluded	Engaging Hill-Sachs excluded	Multidirectional instability excluded
Kim et al. [23]	Arthroscopic Bankart repair: 15 Transglenoid suture: 10 Suture anchor: 5 Open Bankart repair: 8 Transosseous suture: 5 Suture anchor: 3	21 (11-31)	NA	> 30% excluded	NA	Multidirectional or posterior instability or 3° sulcus sign excluded
Krueger et al. [24]	Arthroscopic Bankart repair: 15 Open Bankart repair: 5	NA	NA	> 25% excluded	NA	No Exclusion
Millar et al. [25]	Open procedure: 10 Putti-Platt procedure: 5 Open Bankart repair: 3 Open capsular shift: 2	97 (28–240)	121 (18–264)	NA	NA	Multidirectional or posterior instability excluded
Neri et al. [26]	Arthroscopic Bankart repair: 6 Suture anchor: 5 Transglenoid: 1 Open Bankart repair: 6 Suture anchor: 5 Staple: 1	28 (6–84)	52.5 (9–204)	> 30% excluded	Engaging Hill-Sachs excluded	Multidirectional instability excluded
Patel et al. [27]	Arthroscopic procedure: 21 Arthroscopic Bankart repair: 19 Arthroscopic Bankart repair with posterior labrum repair: 1 Capsular shrinkage: 1 Open procedure: 18 Open Bankart repair: 16 Magnusson-Stack procedure: 2 Unknown: 1	NA	72 (5–308)	Inverted-pear Bankart lesion excluded	Engaging Hill-Sachs excluded	NA
Ryu et al. [28]	Arthroscopic procedure: 11 Arthroscopic Bankart repair: 11 Open procedure: 4 Open Bankart repair: 3 Capsular shift: 1	NA	NA	NA	NA	NA
Shin et al. [29]	NA	NA	NA	> 25% excluded	NA	NA
Total	Arthroscopic procedures: 172/339 (50.7%) Open procedures: 87/339 (25.7%) Unknown: 80/339 (23.6%)	28.09	52.17	-	-	-

NA, Not available.

**Table 4 jcm-09-03418-t004:** Concomitant procedure and complications after revision surgery.

Authors	Concomitant Procedure	Recurrence of Instability	Other Complications	Mechanism of Failure	Re-RevisionRate
Arce et al. [16]	Rotator interval closure: 16/16 Posteroinferior capsular plication: 14/16	3/16 (18.8%)	-	1 T D and 2 NA	1/3
Balazs et al. [17]	SLAP repair: 1/16 SLAP debridement: 9/16 PASTA debridement: 2/16	0/16 (0.0%)	-	0	0/0
Barnes et al. [18]	Rotator interval closure: 1/17 Posterior capsular plication: 13/17 SLAP repair: 3/17	4/17 (23.5%)	-	1 T D and 3 P	1/4
Bartl et al. [19]	Rotator interval closure: 16/56	6/56 (10.7%)	Shoulder stiffness: 2/56 Loosed titanium anchor: 1/56	4 T D 2 A S	4/6
Buckup et al. [20]	NA	3/23 (13.0%)	-	3 T D	3/3
Creighton et al. [21]	Rotator interval closure: 15/18	3/18 (16.7%)	-	1 T D and 2 T S	1/3
Franceschi et al. [22]	Rotator interval closure: 7/10	1/10 (10.0%)	-	1 P	0/1
Kim et al. [23]	Rotator interval closure: 15/23	5/23 (21.7%)	Transient neurapraxia: 1/23	1 T D,2 T S, and 2 P	NA
Krueger et al. [24]	Posterior capsular plication: 1/20	2/20 (10.0%)	Mild osteoarthritis grade I: 5/20	2 P	0/2
Millar et al. [25]	NA	2/10 (20.0%)	-	2 T D	0/2
Neri et al. [26]	Rotator interval closure: 4/12	3/12 (25.0%)	-	1 T D 1 A D 1 T S	NA
Patel et al. [27]	NA	4/40 (10.0%)	-	4 T D	2/4
Ryu et al. [28]	Rotator interval closure: 6/15	4/15 (26.7%)	-	2 T D and 2 A D	2/4
Shin et al. [29]	Rotator interval closure: 7/63	12/63 (19.0%)	-	7 T D,3 A D, and 2 A S	8/12
Total	Rotator interval closure: 102 Capsular plication: 28 SLAP repair: 4 SLAP debridement: 9 PASTA debridement: 2	52/339 (15.3%)	9/339 (2.7%) Stiffness: 2 Transient neurapraxia: 1 Metal failure: 1 Arthritic change: 5	T D: 27/52 (51.9%) T S: 5/52 (9.6%) A D: 6/52 (55.5%) A S: 4/52 (7.7%) P: 8/52 (15.4%) Unknown: 2/52 (3.9%)	22/339 (6.5%)

NA, Not available; SLAP, Superior labrum anterior and posterior; PASTA, Partial articular supraspinatus tendon avulsion; T, Traumatic; D, Dislocation; S, Subluxation; A, Atraumatic; P, Positive apprehension sign; NA, Not available.

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
