# Peer review of "Revision Arthroscopic Bankart Repair: A Systematic Review of Clinical Outcomes"

_jcm, 2020, doi:10.3390/jcm9113418_

Round 1
Reviewer 1 Report
This work despite giving not primary and significant informations to the surgeon, it is concise and well written. Moreover it tries to evaluate and describe a controversial topic and could be helpful to have a better understanding of the pathology and complications related to shoulder instability surgery.
it is well organized and comprehensively described. The english used is readable and correct.
Author Response
Reviewer 1
This work despite giving not primary and significant informations to the surgeon, it is concise and well written. Moreover it tries to evaluate and describe a controversial topic and could be helpful to have a better understanding of the pathology and complications related to shoulder instability surgery.
it is well organized and comprehensively described. The english used is readable and correct.
Response: Thank you for your comment.
Reviewer 2 Report
This is a Systematic review evaluating the clinical outcomes of Arthroscopic revision bankart repair. This paper is precise and well written. Tables and Other files added are well structured. No english editing is needed.
I think it is ready for submission
Author Response
This is a Systematic review evaluating the clinical outcomes of Arthroscopic revision bankart repair. This paper is precise and well written. Tables and Other files added are well structured. No english editing is needed.
I think it is ready for submission
Response: Thank you for your comment.
Reviewer 3 Report
The authors present a systematic review on revision arthroscopic bankart repair and have included 14 studies with a total of 339 shoulders in 337 patients.
The manuscript is comprehensive, well structured and well written.
However, I recommend to add a section on surgical alternatives (e.g. Latarjet procedure) within the discussion.
Detailed Comments:
The authors present a systematic review on revision arthroscopic Bankart repair and have included 14 studies with a total of 339 shoulders in 337 patients. The authors found a recurrent instability rate of 15% and a re-revision rate of 6.5%. The manuscript is comprehensive, well structured and well written. Furthermore, the authors have extracted and documented all relevant aspects (e.g. additional procedures like closure of the rotator cuff interval, chondral defects, e.g.) from the available literature. However, I recommend to add a section on surgical alternatives to a revision Bankart repair (e.g. Latarjet procedure) within the discussion as a revision Bankart repair (which is the actual topic of the systematic review) can also be seen very critical and quite a few surgeons opt for a straight Latarjet procedure after a failed Bankart repair.
Overall, I recommend to accept this manuscript as it provides an interesting and relevant overview - once the procedure is put in sufficient context to other surgical options after failed arthroscopic Bankart repair.
Author Response
Response: Thank you for your comment. According to your suggestion, we added the sentences about surgical alternatives in the third paragraph of the discussion.
“Various procedures emphasizing glenoid reconstitution have become the gold standard in patients with recurrent anterior shoulder instability associated with the bony glenoid defect. Among several surgical methods, the Latarjet procedure has been relatively successful for preventing recurrent dislocation and subluxation, particularly in patients with large glenoid bone loss or failed previous stabilization surgery. In the long-term result, Hurley et al. reported the Latarjet procedure at a minimum of 10 years of follow-up resulted in excellent functional outcomes and a high rate of return to sport among athletes. However, the Latarjet procedure is thought to have more complications, including infection, graft non-union, graft lysis, graft fracture or neurovascular injury with arthroscopic shoulder stabilization procedures.”
